# Expanding the Clinical and Genetic Spectrum of *RAB28*-Related Cone-Rod Dystrophy: Pathogenicity of Novel Variants in Italian Families

**DOI:** 10.3390/ijms22010381

**Published:** 2020-12-31

**Authors:** Giancarlo Iarossi, Valerio Marino, Paolo Enrico Maltese, Leonardo Colombo, Fabiana D’Esposito, Elena Manara, Kristjana Dhuli, Antonio Mattia Modarelli, Gilda Cennamo, Adriano Magli, Daniele Dell’Orco, Matteo Bertelli

**Affiliations:** 1Department of Ophthalmology, Bambino Gesù Children’s Hospital, 00165 Rome, Italy; giancarlo.iarossi@opbg.net; 2Department of Neurosciences, Biomedicine and Movement Sciences, Section of Biological Chemistry, University of Verona, 37134 Verona, Italy; valerio.marino@univr.it; 3MAGI’S Lab s.r.l., 38068 Rovereto (TN), Italy; paolo.maltese@assomagi.org (P.E.M.); matteo.bertelli@assomagi.org (M.B.); 4Department of Ophthalmology, San Paolo Hospital, University of Milan, 20142 Milano, Italy; leonardo.colombo.82@gmail.com (L.C.); antoniomattia.modarelli@gmail.com (A.M.M.); 5Imperial College Ophthalmic Research Unit, Western Eye Hospital, Imperial College Healthcare NHS Trust, London NW1 5QH, UK; f.desposito@imperial.ac.uk; 6MAGI Euregio, 39100 Bolzano, Italy; elena.manara@assomagi.org (E.M.); kristjana.dhuli@assomagi.org (K.D.); 7Eye Clinic, Department of Neurosciences, Reproductive Sciences and Dentistry, University of Naples “Federico II”, 80138 Naples, Italy; 8Eye Clinic, Department of Public Health, University of Naples “Federico II”, 80138 Naples, Italy; xgilda@gmail.com; 9Department of Pediatric Ophthalmology, University of Salerno, 84084 Fisciano (SA), Italy; magli@unina.it

**Keywords:** autosomal recessive cone-rod dystrophy, compound heterozygosis, GTPase, molecular dynamics

## Abstract

The small Ras-related GTPase Rab-28 is highly expressed in photoreceptor cells, where it possibly participates in membrane trafficking. To date, six alterations in the *RAB28* gene have been associated with autosomal recessive cone-rod dystrophies. Confirmed variants include splicing variants, missense and nonsense mutations. Here, we present a thorough phenotypical and genotypical characterization of five individuals belonging to four Italian families, constituting the largest cohort of *RAB28* patients reported in literature to date. All probands displayed similar clinical phenotype consisting of photophobia, decreased visual acuity, central outer retinal thinning, and impaired color vision. By sequencing the four probands, we identified: a novel homozygous splicing variant; two novel nonsense variants in homozygosis; a novel missense variant in compound heterozygous state with a previously reported nonsense variant. Exhaustive molecular dynamics simulations of the missense variant p.(Thr26Asn) in both its active and inactive states revealed an allosteric structural mechanism that impairs the binding of Mg^2+^, thus decreasing the affinity for GTP. The impaired GTP-GDP exchange ultimately locks Rab-28 in a GDP-bound inactive state. The loss-of-function mutation p.(Thr26Asn) was present in a compound heterozygosis with the nonsense variant p.(Arg137*), which does not cause mRNA-mediated decay, but is rather likely degraded due to its incomplete folding. The frameshift p.(Thr26Valfs4*) and nonsense p.(Leu13*) and p.(Trp107*) variants, if translated, would lack several key structural components necessary for the correct functioning of the encoded protein.

## 1. Introduction

Cone-rod dystrophies (CORDs) constitute a genetically heterogeneous group of progressive inherited retinal diseases characterized by a primary loss of cones with a subsequent loss of rods [1]. Clinical hallmarks of CORDs are poor visual acuity, impaired color vision and photophobia, the subsequent impairment of night vision, and peripheral vision loss can occur at a later stage because of rod involvement [1]. Phenotype is characterized by an early onset and progressive macular degeneration with a fundus appearance varying from normal in early stages to central neuroretinal atrophy.

*RAB28*, which encodes a member of the Rab subfamily of the RAS-related small GTPases, is a gene recently associated with autosomal recessive (ar) forms of CORDs [2,3,4,5]. Rab-28 is localized in the cone basal body, in the ciliary rootlet and in the retinal pigment epithelium (RPE), and it is regulated by the guanine nucleotide exchange factors (GEF) myotubularin-related protein 5 [6] and 13 [6,7] belonging to the DENN (differentially expressed in normal and neoplastic cells) domain proteins [8]. The exact molecular mechanism triggering cell death associated with CORD is still unclear. A study on a murine model showed that Rab-28 plays an essential role in cone-specific disc shedding and phagocytosis. This led to the proposition of an impaired membrane shedding at the distal cone outer segment (COS) and/or failed phagocytosis by the RPE as the possible pathogenetic mechanism for CORD [9]. Such evidence has been confirmed in a recent paper reporting altered COS shedding in the zebrafish knockout model [10]. To date, only six variants have been reported worldwide, confirming the extreme rarity of *RAB28*-related CORD. Roosing et al. [2] first reported homozygous nonsense variants in *RAB28* as a cause of arCORD in a German family and a Moroccan Jewish family. Riveiro-Álvarez et al. [3] reported homozygous variants in *RAB28* in two Spanish families: a splice site variant and a missense variant p.(Cys217Trp). Lee et al. [4] described a new homozygous missense *RAB28* variant p.(Ser23Phe) in a Korean patient. Recently, a homozygous missense variant p.(Gly19Arg) has been described in two brothers presenting CORD, myopia, and postaxial polydactyly [5].

In the present study, we analyzed a cohort of five Italian patients, representing the largest cohort of *RAB28* patients clinically and genetically characterized so far. We identified two novel homozygous nonsense variants, a novel homozygous splicing variant and a novel missense variant in compound heterozygosis with a previously reported nonsense variant. In agreement with previous cases reported in the literature, the clinical phenotype was characterized by photophobia, impaired color vision, moderate to severe decrease of visual acuity, and central outer retinal thinning progressing to atrophy. The structural and functional effects of such mutations were evaluated by a thorough analysis based on molecular modeling and molecular dynamics (MD) simulations, which suggested peculiar pathological mechanisms. Our data suggest that the nonsense and frame shift homozygous variants lead to CORD due to complete loss of function, whereas the missense variant in heterozygous compound state results in a partial loss-of-function due to impaired Mg^2+^ and GTP binding.

## 2. Results

All patients reported a poor vision acuity usually from childhood, photophobia, and color vision impairment. Best corrected visual acuity (BCVA) ranged from 1.7 to 0.52 logarithm of the minimum angle of resolution (LogMAR). All patients showed a myopic defect ranging from −2 to −9 D with an associated astigmatic defect ranging from −1.75 to 3.50 D (Table 1). Flash electroretinogram (ERG) recording showed a non-detectable response for the photopic component and a reduced response for the scotopic component. None of the patients reported night blindness despite the reduced scotopic response. Analysis of clinical features of patients were compatible with the diagnosis of cone-rod dystrophy. Genetic testing revealed five variants presenting in homozygous state in four patients and in compound heterozygosis in one, most of which are novel (Table 2), with autosomal recessive inheritance pattern.

### 2.1. Patient 1

The proband is a 17-year-old boy suffering from poor vision since the age of 10. BCVA was 0.52 LogMAR in both eyes with a refractive error of −5.50 sph −1.00 cyl/100° D in RE and of −5.00 sph −2.00 cyl/65° D in LE. The fundus examination showed normal optic disk, a loss of the foveal reflex, and “salt and pepper” peripheral retinopathy. Fundus autofluorescence (FAF) revealed a central hyperfluorescent area in both eyes while optical coherence tomography (OCT) imaging showed a marked reduction of the foveal thickness associated with a photoreceptor layer disruption (Figure 1).

Genetic testing revealed a new homozygous splicing variant that was confirmed by a family segregation study. The homozygous c.76-9A > G variant is located in the branch site in intron 1. The change is predicted to modify the splicing of the mRNA, leading to the retention of a 8nt sequence in the mRNA. Consequently, by conceptual translation, this results in a nonfunctional protein. The *RAB28* gene encodes for 3 isoforms with alternative C termini, all of them carried the c.76-9A > G variant.

We extracted RNA from blood, reverse transcribed it to cDNA, and amplified the sequence of *RAB28* (exons 1 and 2) using a pair of primers (see Methods). We then sequenced the amplicon in the proband and in a control sample and confirmed on mRNA the defective splicing in the proband with the retention in the coding sequence of the TTTTTTAG sequence. The predicted mRNA, if translated, generates a premature stop codon after 30 amino acids, p.(Thr26Valfs4*) (the wild-type protein is of 220 amino acids in length) (Figure 2) and it is likely to undergo either mRNA-mediated decay or protein degradation.

### 2.2. Patient 2

A 33-year-old male proband presented disease onset at the age of 6, with visual acuity reduction, photophobia, and color vision alteration. BVCA decreased progressively over years to 1.00 LogMAR in both eyes, remaining stable for the last six years. Refractive error was −4.25 sph −2.00 cyl/180° in RE and –4.50 sph −1.50 cyl/30° in LE.

The fundus examination showed normal optic disk, signs of foveal atrophy, and “salt and pepper” peripheral retinopathy. Fundus autofluorescence revealed a dark central area due to the absence of the RPE surrounded by an unusual hyperfluorescent perifoveal ring in both eyes while infrared retinography showed foveal atrophy. OCT imaging revealed a marked reduction of the foveal thickness associated with a photoreceptor layer disruption (Figure 3). Goldmann visual field test showed central scotoma with a mild peripheral constriction.

Genetic testing revealed that the proband carried the unpublished nonsense variant p.(Leu13*) in homozygous state, as confirmed by the family segregation study (Table 2). The variant generates a premature stop codon after 13 amino acids and is likely to cause mRNA-mediated decay or protein degradation.

### 2.3. Patient 3

The 38-year-old subject is the sister of patient 2. Despite having the same genotype as the brother, her phenotype showed a milder expression. Uncorrected visual acuity was 0.52 LogMAR after LASIK treatment for the correction of a 9 D myopic defect in both eyes. Ophthalmic features resembled the alterations observed in the brother with a less pronounced foveal atrophy as shown in Figure 4. Differently from the brother, FAF shows a foveal hypofluorescence surrounded by a hyperfluorescent area, indicating the accumulation of fluorophores in RPE due to the defective phagocytosis and incomplete digestion of discs but a more preserved central RPE. It can be noted from OCT imaging that there is a reduction of the foveal thickness with a photoreceptor layer disruption and relative sparing of more peripheral areas. Visual field test showed a relative central scotoma with preserved peripheral extension.

### 2.4. Patient 4

The 64-year-old female proband presented a relative late onset of the disease, reporting the first symptoms of vision impairment around the age of 30. BVCA decreased progressively over the next ~34 years to 1.7 LogMAR in both eyes. Refractive error was −2.00 sph in both eyes. The fundus examination showed a tigroid aspect with vessel thinning, macular atrophy, and optic pallor. Flash full-field ERG recordings showed a non-detectable response for the photopic component and a reduced response for the scotopic response. MfERG response was markedly reduced but still detectable in both eyes, presenting a residual foveal cone function. OCT macular scan showed a marked foveal thinning with advanced photo-receptor layer disruption (Figure 5).

Genetic testing revealed a novel homozygous nonsense variant p.(Trp107*). Family segregation study could not be assessed due to the death of the parents and the absence of other potentially informative family members (Table 2).

### 2.5. Patient 5

The 16-year-old male patient reported poor vision and photophobia since the age of 4. BCVA was 0.4 LogMAR in RE and 0.52 LogMAR in LE with a refractive error of −5.75 sph −3.00 cyl/15° D in RE and of −6.00 sph −3.50 cyl/170° D in LE. The fundus examination showed normal optic disk, a loss of the foveal reflex, and peripheral RPE mottling. Fundus autofluorescence revealed a central hypofluorescence with a mild perifoveal hyperfluorescent area in both eyes while OCT imaging showed a foveal photoreceptor disruption associated with a normal central macular thickness and relative sparing of paracentral retina (Figure 6). Flash full-field ERG recordings showed reduced responses mostly for the photopic component. Visual fields display a central scotoma.

Genetic testing revealed that the proband carried the known nonsense variant p.(Arg137*) [2] and the novel p.(Thr26Asn) missense variant in compound heterozygous state, as confirmed by the family segregation study (Table 2).

### 2.6. Structural and Functional Analysis of Rab-28 Protein

The Ras superfamily of small GTPases can be divided into several families with specific sequence, structure, and cellular function. Among the largest families of Ras GTPases, the Rab family is composed of at least 60 different members specifically involved in membrane trafficking [11], sharing only 30% identical amino acids with all members of the Ras superfamily and with other monomeric GTPases in general.

The overall folding of Rab family proteins consists of six β-strands (named β1 to β6) surrounded by five α-helices (named α1 to α5) forming an α/β fold, where the secondary structure elements are arranged in an incomplete β-barrel fashion, with the missing β-strand substituted by the C-terminal α-helix (α5, Figure 7A). The functional elements responsible for Mg^2+^ and guanine nucleotide binding, as well as for GTP hydrolysis, are represented by the five loops connecting β1-α1 (P-loop), α1-β2 (Switch 1), β3-α2-β4 (Switch 2) and β5-β6-α5 (Figure 7A). Specifically, Switch 1 and Switch 2 regions are involved in a loaded-spring conformational change [12] upon GTP-GDP exchange, consisting of the relaxation of such loops after γ-phosphate release subsequent to GTP hydrolysis. On the other hand, the P-loop is the region involved in phosphate binding via the highly conserved “G motif” (GxxxGKS/T), where the last residue (T26 in Rab-28) is responsible for Mg^2+^ coordination [13]. 

Despite sharing the overall folding and function, phylogenetic analysis of Rab-28 highlights a low sequence identity (<31%) compared with other Rab GTPases, significantly lower than the average (>40%) [14].

For such reasons, it is not surprising that the conformational change between the active (GTP-bound) and the inactive (GDP-3′P, from now G3D, see Methods section) bound form is considerably larger compared to the rest of the Rab family [15]. Indeed, while the P-loop seems substantially unaltered between the active and the inactive form, the Switch 1 region opens up significantly upon GTP-G3D-exchange, as a result of a shortening of β2 strand at the level of Leu 47, in line with the loaded-spring mechanism described for Rab proteins.

### 2.7. Molecular Modeling and Molecular Dynamics Simulations Analysis of the Variants

To evaluate the effects of the missense variant found in patient 5 p.(Thr26Asn) (from now on T26N) on the structural rearrangement subsequent to GTP-G3D exchange, we ran exhaustive 500 ns molecular dynamics (MD) simulations on both the wild-type (WT) and the T26N variant. To assess the stability of the three-dimensional model throughout the simulated timeframe, we calculated the root-mean square deviation (RMSD) of the Cα with respect to the equilibrated structure after 4 ns. Results summarized in Figure 7B showed that both the GTP-bound and the G3D-bound forms of WT Rab-28 were already structurally stable after the first 100 ns of MD simulations, with RMSD values attesting around ~1.5 and ~2.5 Å, respectively. On the other hand, both the active and inactive forms of the T26N variant exhibited higher fluctuations even after 250 ns, with RMSD values approaching ~3.5 and ~4 Å, respectively.

The analysis of the structural flexibility by means of the root-mean square fluctuation (RMSF) calculated on Cα atoms (Figure 7C) highlighted that not only the G3D-bound WT, but also both the active and inactive forms of the T26N variant were substantially more flexible than the GTP-bound WT. In particular, the entire Switch 1 region was found to be considerably more flexible in both the WT and the T26N inactive forms, while the GTP-bound variant displayed decreased structural stability only in the C-terminal part of switch 1 (residues 40–49). Moreover, the increase in flexibility was mutant-specific also in the switch 2 region, as well as in the region encompassing the β5-α4 loop, the α4 helix, and the β6 strand. Finally, the N-terminal part of the α3 helix exhibited a 2-to-3-fold increase in flexibility in the inactive WT and in both active and inactive forms of the variant. It is noteworthy that while the enhanced plasticity of the Switch 1 region was somehow expected, due to the proximity of the T26N variant, such increase in all other identified regions pointed towards an allosteric effect of the missense variant, ultimately leading to an altered interaction with both the guanine nucleotide and the Mg^2+^ ion.

Indeed, while in the GTP-bound form of WT Rab-28, the nucleotide was substantially “locked” between the P-loop and the Switch 1 region, with a solvent accessible surface area (SASA) around 100–120 Å^2^, the inactive form of the WT exhibited a doubled SASA (Figure 7D), with a behavior similar to that of the inactive T26N variant. On the contrary, the active form of the T26N variant displayed an increasing SASA of the GTP throughout the simulation, settling to ~190 Å^2^ after 320 ns, indicative of a looser interaction between the nucleotide and the protein. The RMSF analysis on Mg^2+^ ion yielded comparable results with the GTP/G3D SASA analysis, as the cation was found to be less tightly bound in the inactive form with respect to the active form both in the WT (0.38 vs. 0.59 Å) and in the T26N variant (0.59 vs. 0.77 Å). Interestingly, Mg^2+^ showed 1.5-fold higher structural fluctuations in either forms of the T26N variant compared to their WT counterparts, suggesting a rearrangement in the network of electrostatic interactions ultimately responsible for Mg^2+^-coordination.

A more in-depth analysis of the persistence of electrostatic interactions along the 500 ns MD trajectory highlighted the crucial role of residue 26 in maintaining the correct architecture for Mg^2+^-binding.

As a matter of fact, residue T26 was found be involved in a persistent sidechain H-bond with D68 in both the active and inactive forms of WT Rab-28 (88.3 and 90.1% persistence, respectively, Table 3). Residue D68 belongs to the conserved DxxGQ motif located in Switch 2 region, which is fundamental for the catalytic activity, as the Asp residue stabilizes Mg^2+^, the Gly residue contacts the γ-phosphate, while the Gln acts as catalytic residue for GTP hydrolysis [16]. The substitution in the GTP-bound form of the T26N variant not only significantly decreased the persistence with residue D68 (51.7 vs. 88.3%, Table 3), but also gave rise to a new H-bond with D48, detected for 34.2% of simulation time, both of which were surprisingly absent in the inactive form. Such rewiring of the interactions upon T-to-N substitution of residue 26 resulted in an alteration in Mg^2+^-coordination, as the GTP-bound form of the variant exhibited an interaction between D48 and Mg^2+^ occurring 19.3% of the simulated timeframe, in addition to the interaction of the cation with D68 which was present during the entire trajectory also in the WT (100% persistence in both cases).

It is noteworthy that although the simulated timeframe is enough to evaluate the structural rearrangements caused by the mutation, ion dissociation is a process which requires several orders of magnitude more to occur. For this reason, the remarkable difference in persistence of the interactions between the residues involved in Mg^2+^-coordination is again indicative of a higher propensity of the T26N mutant towards a Mg^2+^-free and GDP-bound state. The inactive form of the WT displayed a 100% persistent interaction between D68 and the cation, which was drastically reduced to 0.2% in the T26N variant, thus implying a complete loss of Mg^2+^-coordination.

In summary, our results suggested that the concerted effect of the structural perturbation induced by the T26N variant on the proximal switch 1 region (Figure 8) and on the distant β5- β6 region would affect both GTP-GDP exchange and Mg^2+^-binding, thus classifying the substitution as a partial loss-of-function missense mutation. Overall, our findings are perfectly in line with several previous lines of evidence on Rab family proteins [17], as the T26N mutation affects the last residue of the broadly conserved G motif among GTPases, and it is widely used to mimic a “GDP-locked” state of Rab proteins [10,13,16,17] due to the disruption of Mg^2+^ binding site and the resultant 100-fold decrease in affinity for GTP.

Concerning the nonsense p.(Arg137*) variant, no nonsense-mediated decay of mRNA was previously reported [2] on CORD patients carrying such variant, suggesting potential functional defects as the pathological mechanism. To assess such hypothesis, the structural analysis highlighted that the truncation of the protein sequence at the level of R137 would deprive the protein structure of α4, β6, and α5 (Figure 9A,B, Appendix A) structural elements. The lack of β6 and α5 would most likely prevent the achievement of the correct α/β fold and affect the guanine nucleotide binding pocket, thus resulting in protein degradation.

The structural analysis of the novel nonsense variant p.(Trp107*) suggested that the interruption of the protein sequence after residue W107 would break up helix α3, thus resulting in the absence of several structural elements, namely the C-terminal part of α3, β5, α4, β6, and α5. Analogously to the nonsense p.(Arg137*) variant, a protein missing the key structural elements β5, β6, and α5 would most probably not be able to fold correctly or bind GTP/GDP (Figure 9A–C, Appendix A), and ultimately would be subjected to degradation. Finally, for the same reasons as the previously analyzed nonsense variants, p.(Leu13*) and p.(Thr26Valfs4*) would either undergo nonsense-mediated mRNA decay, as predicted by the online tool Mutation Taster [18], or retain an even smaller fraction of the structural elements required for the folding of the functional Rab-28 and therefore be degraded.

## 3. Discussion

Cone-rod dystrophy (CORD) represents a rare form of inherited retinal diseases affecting 1/30,000 to 1/40,000 individuals [1] with heterogeneous genetic background. Indeed, more than 30 genes have been associated with CORD, but the specific role of each gene is not always clear, such as in the case of *RAB28.* As all other Rab GTPases, Rab-28 is expected to control specific membrane trafficking processes by encoding the information about the state of a specific membrane or membrane domain [19,20]. This is possible as a consequence of Rab-activation by specific GEFs, that promote the release of GDP and the binding of GTP, thus activating the Rab protein [21], which then recruits and/or activates effector proteins exerting cytoskeletal and membrane tethering functions. The hydrolysis of GTP to GDP, either occurring by the intrinsic GTPase activity or enhanced by additional GTPase-activating proteins (GAPs), ends the cycle. While it has been recently found that specific DENN domains activate Rab-28 [6], which could then serve as GEF, there is a clear lack of mechanistic information as to the other possible molecular complexes formed by Rab-28, hence little is known about its general biological function.

Recent studies on *RAB28* knockout mice showed that Rab-28 plays an essential role in cone-specific disc shedding and phagocytosis and an impaired membrane shedding at the distal COS and/or failed phagocytosis by the RPE has been proposed as possible pathogenetic mechanism for CORD [9]. The role of Rab-28 in regulating membrane shedding from cone outer segment tips has been confirmed in a *RAB28* knockout zebrafish model where, however, a longer visual function with no sign of retinal degeneration up to 12 months has been observed [10]. In humans, only six *RAB28* variants have been reported to be associated with CORD. Three missense variants, p.(Ser23Phe) in a Korean female [4], p.(Cys217Trp) in a Spanish female [3], and p.(Gly19Arg) in two Danish brothers [5], respectively; two nonsense variants: p.(Glu189*) and p.(Arg137*) in a German and a Moroccan Jewish family [2], respectively; a splice variant c.172 + 1G *>* C in a Spanish family [3]. The phenotype of these patients presented features in common, showing a progressive macular atrophy associated with a markedly reduced visual acuity, impairment of color vision, intense photophobia since childhood without a history of night blindness and a myopic pattern. The comparison of the predicted three-dimensional structure of the p.(Ser23Phe) variant with wild type *RAB28* protein suggested impaired ligand binding [4].

In this work, we analyzed the clinical phenotype of five Italian patients affected by *RAB28*-associated CORD and identified two novel homozygous nonsense variants, a novel homozygous splicing variant and a novel missense variant in compound heterozygosis with a previously reported variant. The phenotype was mainly similar in all patients but presented different stages of severity. In agreement with the clinical diagnosis of CORD, common symptoms were reduced visual acuity, impairment of color vision, and intense photophobia since childhood, without a history of night blindness. None of the patients presented nystagmus, thereby supporting the diagnosis of non-congenital visual impairment. In FAF imaging, all patients presented a perifoveal ring or a central macular area of relatively increased autofluorescence; such evidence has been noted for bull’s-eye maculopathy, cone-rod dystrophy, cone dystrophy with supernormal rod response, and rod-cone dystrophy [22,23], and is a non-specific manifestation of cone dystrophy that can occur in other forms of retinal degenerations. However, in the case of patients affected by *RAB28*-related CORD, it can be explained by the accumulation of membrane material due to the defective phagocytosis and cone outer segment shedding. FAF macular hypofluorescence observed in more severe/advanced phenotypes can represent the subsequent progression of photoreceptor layer and RPE degeneration.

Despite the common early onset and progressive retinal degeneration, a significant interindividual and intrafamilial variability of the disease was observed. Severity of phenotype was apparently independent of the gene involvement and the site of nucleotide variations. Patients 2 and 3, carrying the same variants, presented similar ophthalmological features but different severity of the disease. Foveal atrophy was less pronounced in the proband’s older sister, resulting in a more preserved visual acuity; similarly, visual field was less altered in the female subject, who also showed a later onset of the disease compared with her brother.

This heterogeneous clinical manifestation may be explained by a variable expression of similar *RAB28* variants: members of the same family may have the same variant and show different severities of the disorder. Moreover, patient 4 presented a later onset of the disease than generally reported for RAB28-associated CORD, while patient 5, though reporting an early onset of the disease and manifesting a significant reduced visual acuity due to the photoreceptor layer defect, still maintained a normal central macular thickness.

Reports on animal models showed a different impact of *RAB28* variants on retinal function among species. Studies on mouse and zebrafish confirmed the role of *RAB28* in cone outer segment shedding mechanism, but in contrast to the mouse *RAB28* knockout [9], *RAB28* knockout zebrafish display decreased RPE phagosomes, but normal visual function up to 21 days post fertilization and no retinal degeneration up to 12 months post fertilization [10]. One explanation of this different impact of *RAB28* variants on retinal function could be that the level of outer segment shedding/phagocytosis remaining in zebrafish *RAB28* mutants is enough to support photoreceptor survival. Alternatively, it has been recently demonstrated that genetic lesions which induce nonsense-mediated decay of mRNA can elicit a compensatory transcriptional response, whereby genes with similar functions are upregulated, masking the effect of the mutant gene [24]. Though conditions among species can vary and be influenced by intrinsic factors (i.e., cellular density, growth, and regeneration), it cannot be excluded that regulatory factors may play a role in the clinical expressivity of *RAB28* variants in humans. In agreement with the literature, all our patients showed a myopic pattern ranging from −2 to −9 diopters; though this ophthalmic feature seems to be a distinctive factor associated with *RAB28*-related CORD, probably more cases are necessary to speculate whether *RAB28* variants may cause refractive errors as well as retinal degeneration.

From a molecular standpoint, the structural analysis of the nonsense and splicing variants here identified suggested that the lack of several key structural elements would either lead to protein degradation or mRNA-mediated decay. As far as patient 5 is concerned, the missense variant p.(Thr26Gln) found in compound heterozygosis affects one of the most highly conserved residues of the P-loop, resulting in a partial loss-of function due to the impaired Mg^2+^-coordination, which locks Rab-28 in a GDP-bound inactive state [10,13,16,17], ultimately leading to the altered membrane trafficking. Interestingly, the P-loop seems to be a hotspot region for CORD-associated missense variants, as also p.(Gly19Arg) [5] and p.(Ser23Phe) [4] belong to the same conserved 7-residues stretch. In order to achieve a complete understanding of the role of *RAB28* in triggering cell death in CORD, further molecular work is needed to identify its specific GEFs and GAPs, which play an essential role in its activation and inactivation.

## 4. Materials and Methods

### 4.1. Patient Studies, Clinical and Ophthalmological Examinations

All procedures in this study adhered to the tenets of the Declaration of Helsinki and were approved on 18 November 2020 by the Ethical Committee of Azienda Sanitaria dell’Alto Adige, Italy (Approval No. 132-2020). Probands (age ranging from 16 to 78 years) and their families were examined in different eye clinics. They underwent comprehensive age-appropriate ophthalmic examination, including best corrected visual acuity (BCVA) measurement with the Early Treatment Diabetic Retinopathy Study (ETDRS) charts, expressed as a logarithm of the minimum angle of resolution (logMAR), slit-lamp biomicroscopy, indirect ophthalmoscopy with +90 D noncontact lens (Volk), fundus autofluorescence (FAF), OCT imaging, and full-field electroretinogram (ERG) and multifocal electroretinogram (MfERG) recorded according to the ISCEV standards. Refractive errors ranged between −2 and −9 D spherical equivalent. All examinations were performed at the referring clinic, and the results were collected at the end of the study. All patients received genetic counseling to explain the risks and benefits of genetic testing and gave their informed consent. Demographic and clinical data of patients are shown in Table 1.

### 4.2. Genetic Testing

Genetic testing was performed at MAGI’s Laboratories (MAGI’S Lab, Rovereto and MAGI Euregio, Bolzano, Italy) where patients’ blood or saliva were sent with the clinical diagnosis of CORD.

DNA was extracted with a commercial kit (Blood DNA Kit E.Z.N.A.; Omega Bio-Tek Inc., Norcross, GA, USA) and analyzed using a MiSeq personal sequencer (Illumina, San Diego, CA). The CORD gene-targeted panel comprises of the following genes: *ABCA4*, *ADAM9*, *AIPL1*, *C8orf37*, *CFAP410* (*C21orf2*), *CACNA1F*, *CACNA2D4*, *CDHR1*, *CEP78*, *CNGA3*, *CRX*, *DRAM2*, *GUCA1A*, *GUCY2D*, *UNC119*, *IFT81*, *KCNV2*, *PDE6C*, *PITPNM3*, *POC1B*, *PROM1*, *PRPH2*, *RAB28*, *RAX2*, *RIMS1*, *RPGR*, *RPGRIP1*, *SEMA4A*, *TTLL5*. The segregation studies were performed using Sanger sequencing on a Beckman Coulter CEQ 8000 sequencer (Beckmann Coulter, Milan, Italy). All methods have been published elsewhere [25,26]. All variants were submitted to VarSome for pathogenicity evaluation [27] in accordance with the American College of Medical Genetics and Genomics (ACMG) guidelines [28].

For the evaluation of the splicing variant, total RNA was extracted from blood using the Tempus™ Spin RNA Isolation Kit following manufacturer protocols. The SuperScript VILO cDNA Synthesis Kit was used to generate first strand cDNA. The primers used to obtain the mRNA sequence were AGGACCGGCAACTGAAAATC and TGTTTGTACTGTTTCCCAAAAGT respectively on exon 1 and exon 2 of *RAB28*. The Sanger sequencing reaction was performed as described previously [25] and following manufacturers protocols.

The analysis of the translation of the mutated mRNA was performed by using the tool (https://web.expasy.org/translate/) which allows the translation of a nucleotide (DNA/RNA) sequence to a protein sequence.

### 4.3. Molecular Modeling and Molecular Dynamics Simulations

Molecular dynamics simulations of human Rab-28 in its active (GTP-bound) and inactive (GDP-3′P-bound) forms were set up based on the crystallographic structures with PDB entries 3E5H (resolution 1.5 Å) and 2HXS (resolution 1.1 Å), respectively [15]. As the active form was solved using the non-hydrolyzable GTP analogue GppNHp, the secondary amine NH was substituted with the physiological O restoring the molecule to the physiological GTP-bound form. The inactive form was found to be bound to GDP-3′P instead of the expected GDP [15], but since the protein–nucleotide contacts were indistinguishable, the crystallographic structure was not modified before running simulations. Crystallographic waters within 5 Å of any atom of either the Mg^2+^ ion or the substrate (GTP or GDP-3′P) were retained throughout the simulations. In silico variant T26N was introduced using the most probable non-clashing backbone dependent rotamer proposed by Maestro v. 12.4.072 (Schrodinger) suite.

All-atom MD trajectories were obtained using GROMACS 2019.2 simulation package [29], proteins and ligands were simulated according to CHARMM36m force field [30], parametrization of GTP and GDP-3′P was performed using the input generator module of CHARMM-GUI [31]. System size was approximately 39,500 atoms for all Rab-28 variants, protocols for system preparation and energy minimization were essentially the same as elucidated in [32], the two-step equilibration procedure at constant volume and temperature (310 K) or constant pressure and temperature (1 atm and 310 K, respectively) was performed as detailed in [33]. Finally, wild-type Rab-28 and T26N variant underwent extensive 500 ns MD simulations using the setup detailed in [34].

The molecular mechanics interaction energy and the solvent accessible surface area (SASA) were calculated using the functions *gmx energy* and *gmx sasa* provided by GROMACS package, smoothing of SASA was obtained by calculating the running average over 1 ns trajectory. The root-mean square deviation (RMSD) of Cα with respect to the equilibrated structure after 4 ns was calculated using GROMACS *gmx rms* function. The root-mean square fluctuation (RMSF), that is as the average root-mean square deviation of Cα over the simulated timeframe with respect to the average structure, which represents the local flexibility of the backbone, was calculated by GROMACS function *gmx rmsf*. The calculation of the persistence of H-bonds and electrostatic interactions along the 500 ns trajectories was carried out using PyInteraph [35], using the distance and angle parameters described in [33].

## Figures and Tables

**Figure 1 ijms-22-00381-f001:**
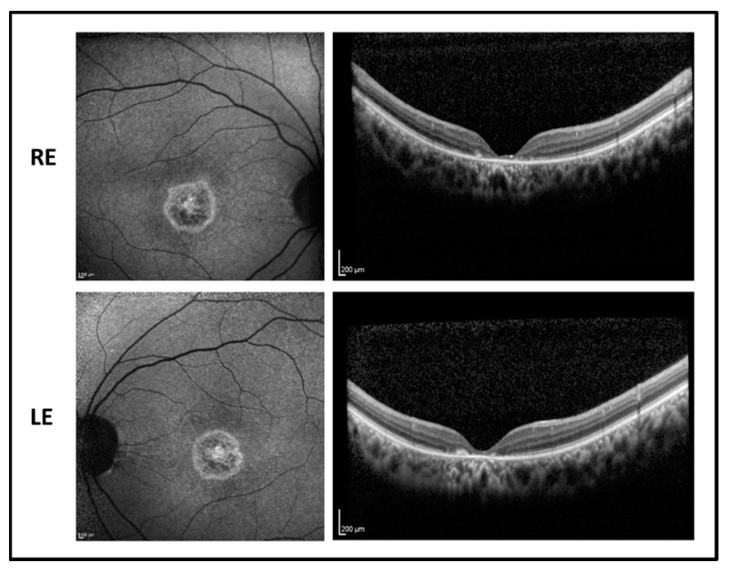
Fundus autofluorescence (**left side**) and optical coherence tomography (OCT) (**right side**) images of the right eye (RE) and left eye (LE) of patient 1 showing macular hyperfluorescence and foveal atrophy with backscattering phenomenon.

**Figure 2 ijms-22-00381-f002:**
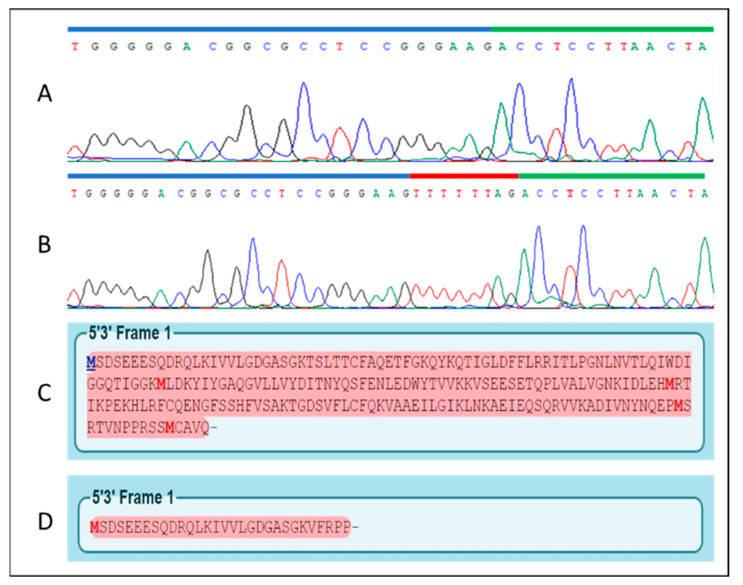
Sequence chromatograms of the wild-type mRNA (**A**) and the c.76-9A > G variant (**B**). The blue bar represents a portion of Exon 1, the green bar a portion of Exon 2, the red bar shows the retention of a 8nt sequence caused by the splicing variant in intron 1. Below, the wild-type amino acid sequence (**C**) and the predicted shorter sequence translated from the mutant mRNA (**D**). Pink shading represents the open reading frame, bold red letters identify Methionine residues or start codons.

**Figure 3 ijms-22-00381-f003:**
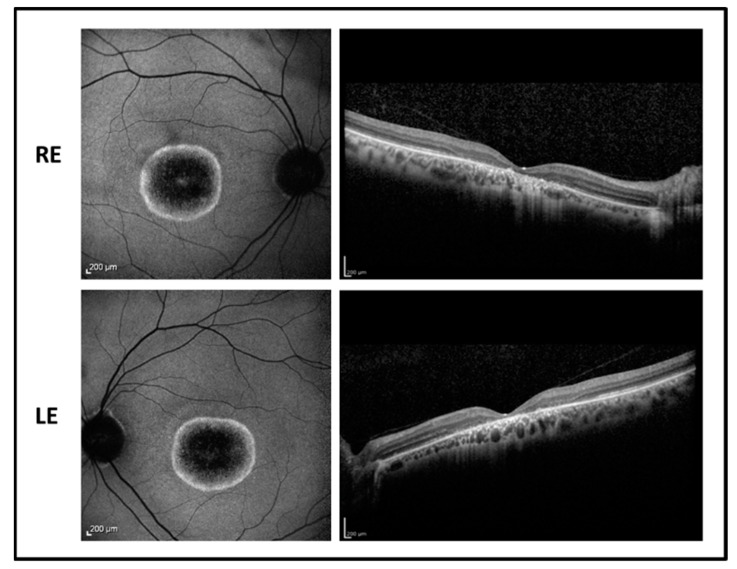
Fundus autofluorescence (**left side**) and OCT (**right side**) images of the right eye (RE) and left eye (LE) of patient 2 showing central hypofluorescence with hyperfluorescent perifoveal ring and marked reduction of foveal thickness with photoreceptor defect.

**Figure 4 ijms-22-00381-f004:**
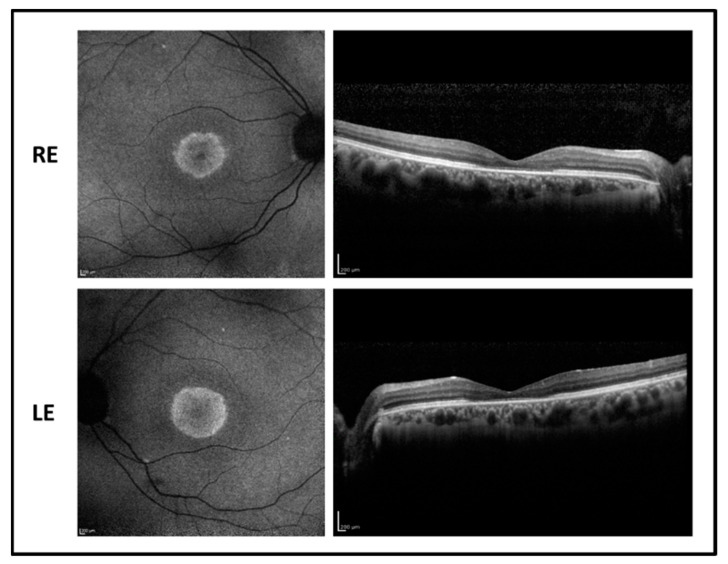
Fundus autofluorescence (**left side**) and OCT (**right side**) images of the right eye (RE) and left eye (LE) of patient 3 showing hyperfluorescent macular area with foveal hypofluorescence and central macular thickness reduction with foveal photoreceptor disruption.

**Figure 5 ijms-22-00381-f005:**
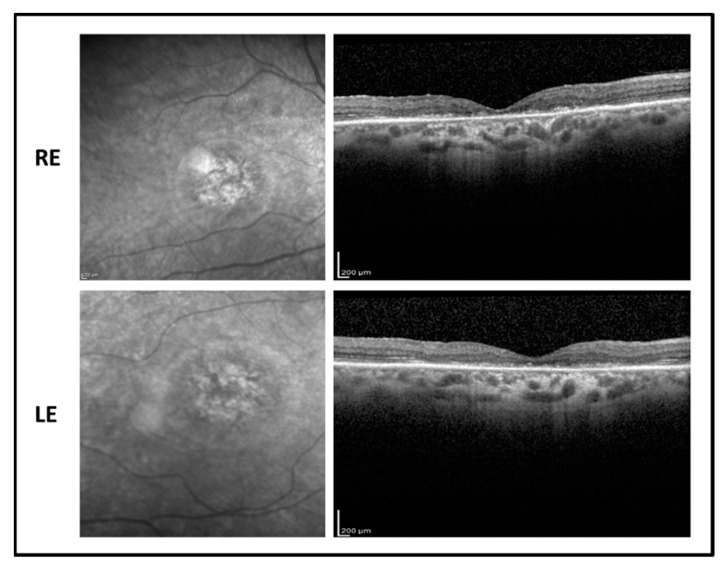
Infrared (**left side**) and OCT (**right side**) images of the right eye (RE) and left eye (LE) of patient 4 showing a marked foveal thinning with advanced photo-receptor layer disruption. Scale bar shown in the upper left panel applies also to lower left panel.

**Figure 6 ijms-22-00381-f006:**
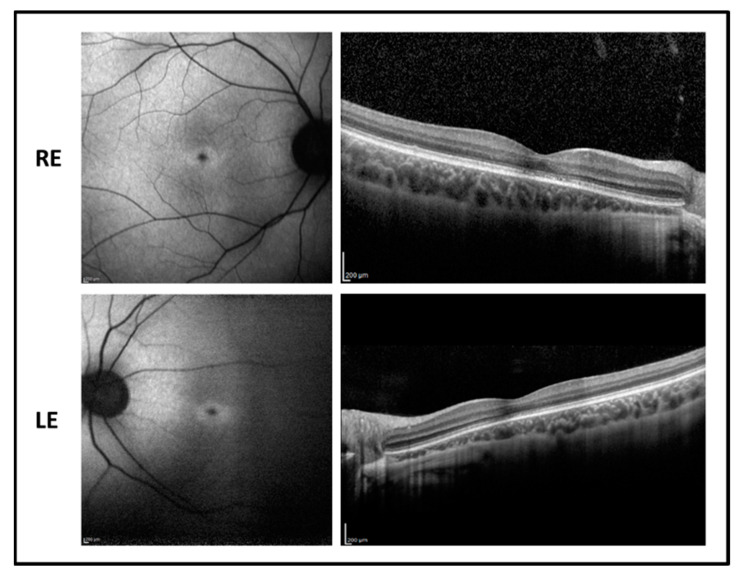
Fundus autofluorescence (**left side**) and OCT (**right side**) images of the right eye (RE) and left eye (LE) of patient 5 showing hyperfluorescent macular area with foveal hypofluorescence and foveal photoreceptor disruption with normal central macular thickness.

**Figure 7 ijms-22-00381-f007:**
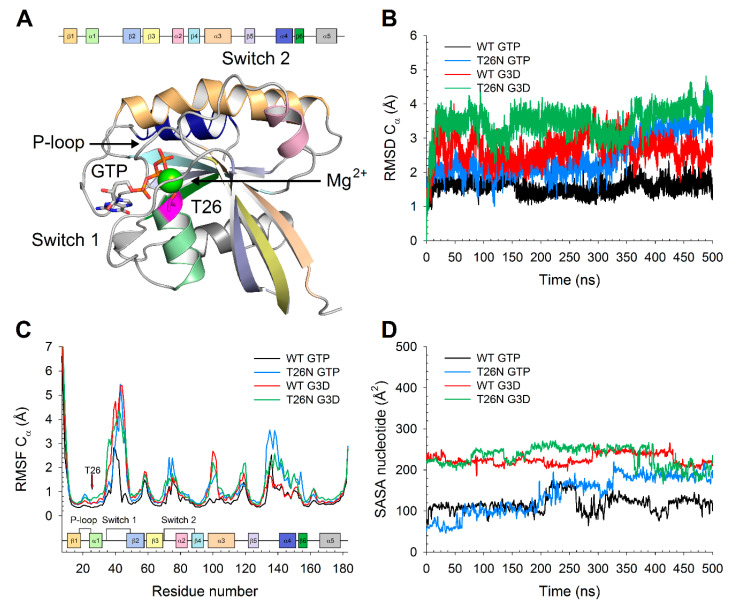
(**A**) The three-dimensional structure of GTP-bound Rab-28 is shown as cartoon, Mg^2+^-ion is shown as a green sphere, GTP is represented as sticks with C atoms in white, N atoms in blue, O atoms in red, and P atoms in orange, residue T26 is shown as magenta sticks with the O atom highlighted in red. The P-loop and the Mg^2+^ ion are indicate by black arrows. Inset shows the coloring scheme for secondary structure elements. (**B**) Time evolution over 500 ns of the Cα-RMSD of GTP-bound wild-type (WT) (black), GTP-bound T26N (blue), G3D-bound WT (red), G3D-bound T26N (green) Rab-28. (**C**) Cα-RMSF profiles of GTP-bound WT (black), GTP-bound T26N (blue), G3D-bound WT (red), G3D-bound T26N (green) Rab-28. Inset shows secondary structure elements colored according to panel A, the position of residue 26 on the secondary structure representation is marked with a red arrow and labeled. (**D**) Time evolution over 500 ns of the SASA of guanine-nucleotide of GTP-bound WT (black), GTP-bound T26N (blue), G3D-bound WT (red), G3D-bound T26N (green) Rab-28. Data was smoothed using a moving average over 1 ns window as described in the methods section.

**Figure 8 ijms-22-00381-f008:**
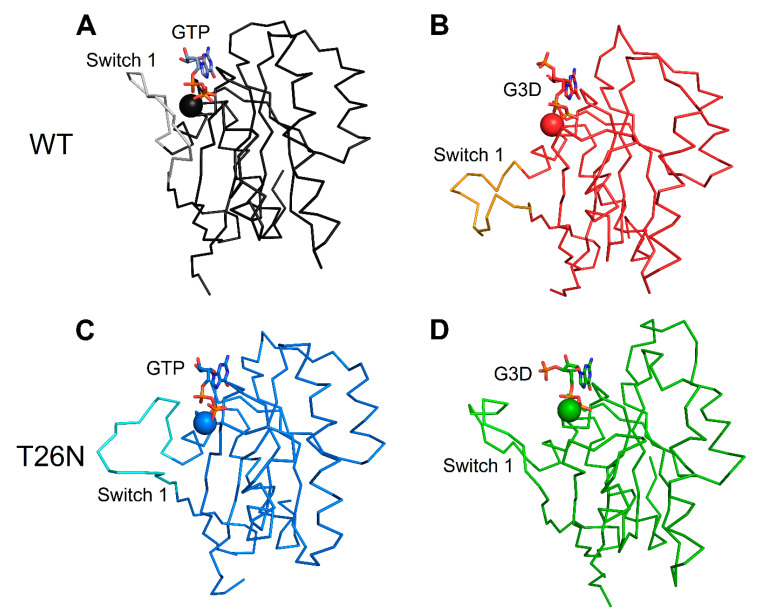
(**A**) Comparison of the three-dimensional structure of (**A**) GTP-bound WT (black), (**B**) GTP-bound T26N (blue), (**C**) G3D-bound WT (red), and (**D**) G3D-bound T26N (green) Rab-28 after 500 ns MD simulations. Protein structure is shown as a ribbon connecting Cα atoms, Mg^2+^-ion is represented as a sphere colored with the same color as the protein structure, GTP/G3D are represented as sticks with N atoms in blue, O atoms in red, and P atoms in orange and labeled. The switch 1 region of each state is highlighted with a lighter hue of the protein structure color and labeled.

**Figure 9 ijms-22-00381-f009:**
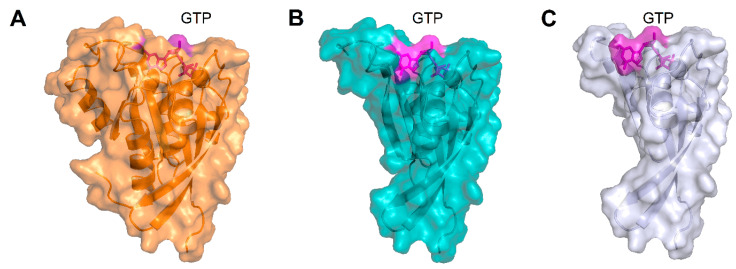
(**A**) Three-dimensional structure of GTP-bound WT Rab-28, protein structure is represented as an orange cartoon, Mg^2+^-ion is shown as an orange sphere, GTP is displayed as magenta sticks, the molecular surface of each element is shown in transparency and colored accordingly. (**B**) Three-dimensional structure of GTP-bound p.(Arg137*) Rab-28, protein structure is represented as a teal cartoon, Mg^2+^-ion is shown as a teal sphere, GTP is displayed as magenta sticks, the molecular surface of each element is shown in transparency and colored accordingly. (**C**) Three-dimensional structure of GTP-bound p.(Trp107*) Rab-28, protein structure is represented as a light blue cartoon, Mg^2+^-ion is shown as a light blue sphere, GTP is displayed as magenta sticks, the molecular surface of each element is shown in transparency and colored accordingly. The GTP molecule shown in panels (**B**,**C**) is merely for representation purposes, as both truncated forms would not allow the correct folding of the binding pocket.

**Table 1 ijms-22-00381-t001:** Demographic and clinical data of patients.

Pt	Sex	Age (Yrs)	BCVA (LogMAR) RE LE		Refractive Error (D)		Onset (Yrs)	OCT	Visual Field
1	M	17	0.52	0.52	−5.50 sph −1.00 cyl/100°	−5.00 sph −2.00 cyl/65°	10	CNA	CS
2	M	33	1	1	−4.25 sph −2.00 cyl/180°	−4.00 sph −2.50 cyl/30°	6	CNA	CS, MPC
3	F	38	0.52	0.52	−9.00 sph	−9.00 sph	10	CNA	CS
4	F	64	1.7	1.7	−2.00 sph	−2.00 sph	30	CNA	CS
5	M	16	0.4	0.52	−8.00 sph −3.00 cyl/15°	−8.50 sph −3.50 cyl/170°	4	CNA	CS

BCVA: best corrected visual acuity; D: diopters; OCT: optical coherence tomography; CNA: central neuroretinal atrophy; CS: central scotoma; MPC: mild peripheral constriction.

**Table 2 ijms-22-00381-t002:** Demographic and genetic characteristics of *RAB28* families. Mutations refer to the following entries in GenBank (https://www.ncbi.nlm.nih.gov/genbank): NG_033891, NM_001017979.3.

Pt	Family Grade	Family Consanguinity	Sex	Age	Affected	Exon/ Intron	Nucleotide Change	Amino Acid Change	Allele State	dbSNP rs	gnomAD MAF ^a^	Classification	Reference
1	Proband	Yes ^b^	M	18	Yes	int1	c.76-9A > G	p.(Thr26Valfs4*)	HOM	rs372774679	ƒ = 0.0000165	VUS	This work
	Mother		F	40	No	int1	c.76-9A > G	p.(Thr26Valfs4*)	HET	rs372774679		VUS	This work
	Father		M	39	No	int1	c.76-9A > G	p.(Thr26Valfs4*)	HET	rs372774679		VUS	This work
2	Proband	No	M	34	Yes	ex1	c.37del	p.(Leu13*)	HOM	rs746362842	ƒ = 0.0000133	Pathogenic	This work
3	Sister		F	40	Yes	ex1	c.37del	p.(Leu13*)	HOM	rs746362842		Pathogenic	This work
	Father		M	68	No	ex1	c.37del	p.(Leu13*)	HET	rs746362842		Pathogenic	This work
4	Proband	Unknown	F	64	Yes	ex4	c.321G > A	p.(Trp107*)	HOM	NA	Not available	Pathogenic	This work
5	Proband	Unknown	M	16	Yes	ex5 ex2	c.409C > T c.77C > A	p.(Arg137*) p.(Thr26Asn)	HET HET	rs398123044 NA	ƒ = 0.00000798	Pathogenic VUS	Roosing et al. [2] This work
	Mother		F	43	No	ex2	c.77C > A	p.(Thr26Asn)	HET	NA	Not available	VUS	This work
	Brother		M	22	No	ex5	c.409C > T	p.(Arg137*)	HET	rs398123044		Pathogenic	Roosing et al. [2]
	Father		M	51	No	ex5	c.409C > T	p.(Arg137*)	HET	rs398123044		Pathogenic	Roosing et al. [2]

^a^ gnomAD minor allele frequency (MAF) from: https://gnomad.broadinstitute.org/. ^b^ parents of Proband 1 are distantly related subjects.

**Table 3 ijms-22-00381-t003:** Persistence over 500 ns molecular dynamics (MD) simulations of the interactions involving T/N26 and Mg^2+^-binding. H-bonds and electrostatic interactions are shown in italics and underlined, respectively.

Variant	GTP-Bound (Active)		G3D-Bound (Inactive)	
	Interaction	Persistence (%)	Interaction	Persistence (%)
**WT**	*T26-D68*	*88.3*	*T26-D68*	*90.1*
	D68-Mg^2+^	100	D68-Mg^2+^	100
**T26N**	*N26-D68*	*51.7*		
	*N26-D48*	*34.2*		
	D48-Mg^2+^	19.3		
	D68-Mg^2+^	100	D68-Mg^2+^	0.2

## Data Availability

The data presented in this study are available on request from the corresponding author. The data are not publicly available due to their size.

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
