# Peer review of "Expanding the Clinical and Genetic Spectrum of RAB28-Related Cone-Rod Dystrophy: Pathogenicity of Novel Variants in Italian Families"

_ijms, 2020, doi:10.3390/ijms22010381_

Round 1

Reviewer 1 Report

In this research article by Iarossi et al, authors have claimed that they identified novel mutations and missense mutation harboring variants of Rab-28 protein, which has been reportedly associated with cone-rod dystrophy, in Italian families. While the study suggests some important clues to how these pathogenic mutations alters the structural properties of Rab28 leading to enhanced mRNA decay and production of pathogenic shorter peptide. However, their investigation lacks enough scientific evidences to support the claim strongly in this manuscript. Please find the suggestions/comments below, which need to be addressed to improve the scientific soundness of this study.

1) Authors mentioned one of the Rab 28 mutation as p.(Thr26Valfs4*) in the Abstract. They should consider mentioning farnesylated abbreviated as "fs" here for the first time. 

2) It is not clear from the text that which are the novel mutations and variants of Rab28 that authors claim as novel finding. They should mention these clearly in the abstract section as well as in appropriate places the manuscript.

In this context, I should mention that although authors claim that T26N fs is a novel finding by their group, however, this mutation has already been reported in a previous study. Likewise other mutations have also been published elsewhere. In that case, authors may consider clearly mentioning the reported and novel mutations with proper citations in Table 2 to eliminate the ambiguity.

3) In Figure 2, authors only predicted the occurrence of pathogenic shorter peptide produced by enhanced mRNA decay of the Rab28 alternative splice variant, which definitely do not provide enough evidence to strongly support their statement. Authors should show by expressing the WT and mutant variants tagged with FLAG or Myc and/or with GFP fusion in appropriate mammalian cell line by western blotting and/or Immunofluorescence that these variants really could undergo altered mRNA decay method.

4) It is not clear from the text which are exactly the alternatively spliced variant. Authors should considering providing a figure with mentioning splice variants including location of the exon-intron boundary(s) involved in this alternative splicing.

5) Authors may consider using some scoring scale to semi-quantitatively analyze the extent of macular degeneration in the subjects, which would greatly help the readers to easily compare the disease stages as well as to relate the pathogenic effects of the respective mutants to a particular disease stage.

Author Response

Reviewer 1

In this research article by Iarossi et al, authors have claimed that they identified novel mutations and missense mutation harboring variants of Rab-28 protein, which has been reportedly associated with cone-rod dystrophy, in Italian families. While the study suggests some important clues to how these pathogenic mutations alters the structural properties of Rab28 leading to enhanced mRNA decay and production of pathogenic shorter peptide. However, their investigation lacks enough scientific evidences to support the claim strongly in this manuscript. Please find the suggestions/comments below, which need to be addressed to improve the scientific soundness of this study.

We are glad that this Reviewer found merit in our work and appreciate her/his suggestions to improve the manuscript. We believe that we already provided strong evidence in support of our conclusions, as it will be made clear in following. We addressed all the points raised by this Reviewer as follows.

1) Authors mentioned one of the Rab 28 mutation as p.(Thr26Valfs4*) in the Abstract. They should consider mentioning farnesylated abbreviated as "fs" here for the first time. 

We believe that the Reviewer probably misinterpreted the “fs” part of the mutation, as it does not stand for farnesylated but it rather stands for “frameshift” and is commonly used in the current nomenclature of genetic variants. However, we made it clearer by rewording the sentence (line 51) in: “The frameshift p.(Thr26Valfs4*) and nonsense p.(Leu13*) …variants….”.

2) It is not clear from the text that which are the novel mutations and variants of Rab28 that authors claim as novel finding. They should mention these clearly in the abstract section as well as in appropriate places the manuscript. In this context, I should mention that although authors claim that T26N fs is a novel finding by their group, however, this mutation has already been reported in a previous study. Likewise other mutations have also been published elsewhere. In that case, authors may consider clearly mentioning the reported and novel mutations with proper citations in Table 2 to eliminate the ambiguity.

We are sorry, but it is not clear to us which variant the reviewer indicated as “T26N fs”. We reported two variants involving the codon #26, namely the p.(Thr26Valfs4*) and the p.(Thr26Asn). However, after extensive check in databases such as HGMD, Varsome, dbSNP and Pubmed, we can confirm that both these RAB28 variants are described in our paper for the first time. In the same context, we also confirm that all variants, with the exception of the p.(Arg137*), are unpublished. Therefore, we added a new column in Table 2 to show unpublished and known variants. We would also like to point that, while we agree with this Reviewer that the T26N variant was reported in previous studies, it was introduced in an animal model to mimic the GDP-bound state of Rab-28 (references 10,13,16,17), but never been found in patients before. For this reason, we do not think that such references would be suitable for Table 2.

3) In Figure 2, authors only predicted the occurrence of pathogenic shorter peptide produced by enhanced mRNA decay of the Rab28 alternative splice variant, which definitely do not provide enough evidence to strongly support their statement. Authors should show by expressing the WT and mutant variants tagged with FLAG or Myc and/or with GFP fusion in appropriate mammalian cell line by western blotting and/or Immunofluorescence that these variants really could undergo altered mRNA decay method.

            We agree with this Reviewer that it would be very interesting to investigate the molecular mechanism behind the p.(Thr26Valfs4*) variant but, in our opinion, this would require a dedicated study as the suggested experiment falls beyond the scope of this manuscript. Nevertheless, we added to lines 137-138 the sentence “and it is likely to undergo either mRNA-mediated decay or protein degradation.” and to lines 180-181 ”or protein degradation” to clarify the speculative nature of the statements.

4) It is not clear from the text which are exactly the alternatively spliced variant. Authors should considering providing a figure with mentioning splice variants including location of the exon-intron boundary(s) involved in this alternative splicing.

            We believe that the only splicing variant found in this study, resulting in the frameshift mutation p.(Thr26Valfs4*), was described in detail in paragraph 2.1, line 156 “The homozygous c.76-9A>G variant is located in the branch site in intron 1.” We modified the figure legend (line 164), including the information on the location of the variant as follows: “… the splicing variant in intron 1. Below, the wild-type amino acid sequence… “

5) Authors may consider using some scoring scale to semi-quantitatively analyze the extent of macular degeneration in the subjects, which would greatly help the readers to easily compare the disease stages as well as to relate the pathogenic effects of the respective mutants to a particular disease stage.

We thank the Reviewer for the suggestion. Correlation between functional /morphological features and variants has been considered but number of patients and not uniform data, coming from different institutions, could not permit a proper classification of the severity of disease. As reported from the analysis of FAF and OCT, we can only suggest that patient 2 shows a more severe stage of macular degeneration than the sister (pt. 3) even if presenting the same mutation, confirming the variability of phenotype expression in RAB28 patients. On the other hand, severity of visual impairment in patient 4 is probably related to her age and length of disease. This is consistent with the few reports in literature showing the need of longer longitudinal studies to better classify a possible scale of severity/progression in CORD through the analysis of different morpho-functional parameters in larger groups of patients.

Reviewer 2 Report

Paper re:  'Novel pathogenic variants in RAB28 in cone-rod dystrophy'

This paper reports 4 novel variants in the RAB28 gene in 5 cases (from 4 families) with autosomal recessive cone-rod dystrophy, and provides evidence, primarily from protein conformation modelling, for their pathogenicity. As one of the rarer causes of cone-rod dystrophy, this confirmation of the involvement of RAB28 is helpful, and should assist with interpretation of any future novel variants in RAB28 found in diagnostic gene panel tests.

The science and the writing of the paper are to a high standard, and revision requires only a few minor improvements to the English, and to the counting of the variants, plus a suggested improvement in the labelling of the colour diagrams.  These are :

1. Abstract lines 39-41, and Introduction lines 82-84, and Discussion lines 444-445.  Four probands should have 8 affected alleles.  The listing of these here (i,ii,iii) suggests there are 9 affected alleles, which is confusing.  In fact there were 3 novel variants in homozygosis, but one of these is the same one as the splice site variant (Case 1), so that cannot be listed as a separate variant.  The authors must amend this sentence accordingly.  Also in this sentence the authors must make it clear that the nonsense variant in compound heterozygosity with the novel missense variant is not novel.  I would suggest rephrasing as ‘a novel missense and a previously reported nonsense variant in compound heterozygosis.’

  1. Introduction lines 82-84 . See Point 1. Above.
  2. Discussion lines 444-445. See Point 1. Above

  3. Intro. line 69. Better as : ‘A study on a murine model…’

  4. Intro. line 71. Better as : ‘…proposed as the possible…’

  5. Intro. line 69-71. Also line 428-431. This whole sentence needs breaking up as it is unclear which aspects are observed and which are proposed.  This could be done by a comma between  ‘…phagocytosis, and….’  (if that is the right place to split), or perhaps better by making 2 sentences.  Ie.  ‘…shedding and phagocytosis.  An impaired membrane shedding at….’

  6. Res. 2.3 line 195-196. Better as : ‘…It can be noted from OCT imaging that there is a reduction…’

  7. Re. 2.4 line 217. Better as : ‘…decreased progressively over the next ~34 years…’

  8. Res.2.4 line 224. ‘…novel…’ may be better than ‘…new…’
  9. Res. 2.5 line 252 ‘…novel…’ may be better than ‘…new…’

  10. Res.2.6 line 271 ‘…peculiar…’ is not the correct word to use here. Do the authors mean : ‘characteristic’, ‘specific’, ‘individual’, or ‘different’ ?

  11. Res.2.6 line 273 ‘…is composed of..’ rather than ‘…is composed by…’

  12. Res.2.6 line 277 ‘…five α-helices (named α1 to α5) named an α/β fold…’ ; the repetition of ‘named’ is confusing. This might be better written as : ‘…five α-helices (named α1 to α5) forming an α/β fold…’

  13. Res.2.6 line 278 ‘…substituted with…’ would read better as : ‘…substituted by…’

  14. Fig7 line 308. Better here to start the sentence with:  ‘A) This cartoon shows…..

  15. Fig.7 line 308-313. It would be much better here to use a colour key in colour, rather than describing the colours. Can this be part of the figure itself; if necessary by making 7A a figure on its own ?

  16. Fig.7 line 313-318. For clarity of reading, could B),C) & D) all start on a new line ?

  17. Molec. Modelling line 324. ‘…Switch 1 region resulted considerably more flexible in both…’   It is difficult to follow what this sentence means.  Do the authors mean : ‘‘…Switch 1 region was found to be considerably more flexible in both…’   Please edit as appropriate.

  18. Molec. Modelling line 329. Better to say: ‘…It is noteworthy that while the enhanced…’

  19. Molec.Modelling line 340. ‘…as the cation resulted less tightly bound…’ Do the authors mean: ‘…as the cation remained less tightly bound…’   or ‘…as the cation was less tightly bound…’  Please change as appropriate.

  20. Fig.8. line 396-412 Please label the GTP in the diagram. Again here, as with Fig7, it would seem better to use a colour key in colour, rather than describing the colours

  21. Disc. Line 428-431. See Point 6. Above (this sentence is repeated from Intro. Line 69-71.

  22. Disc. Line 437-438. This sentence would be better written as : ‘The phenotype of these patients presented features in common, showing a…’

  23. Disc. Line 440-442. The reading of this sentence is ambiguous. Do the authors mean: ‘Only in the case of the p.(Ser23Phe) variant could a three-dimensional structure of the wild type and mutated RAB28 proteins be predicted, and this suggested impaired ligand binding [4]’ ;  or do they mean: ‘Only in the case of the p.(Ser23Phe) variant did a predicted three-dimensional structure of the wild type and mutated RAB28 proteins suggest impaired ligand binding [4]. ?

  24. Disc. Line 450. Better to write: ‘In FAF imaging all patients presented a…’

  25. Disc. Line 452-453 Please add in a comma: ‘…and rod-cone dystrophy [21,22], and is a non-specific…’

  26. Disc. Line 463-464. Which of the sister or brother was the proband ? Here it suggests the sister, whereas in Table 2 it is the brother who is listed as the proband. Please make these consistent.
  1. Disc. Line 464. ‘…disease compared with her brother…’ would seem better than : ‘…disease in respect to the brother…’.

  2. Disc. Line 468. Better with a comma: ‘…CORD, while…’

  3. Disc. Line 473. ‘…in contrast to the mouse…’ would be better than:  ‘…differently from the mouse…’

  4. Disc. Line 496. Better as : ‘…understanding of the role of…’

  5. Abbreviations line 578. Please list the abbreviations in alphabetical order.

Author Response

Reviewer 2

Paper re:  'Novel pathogenic variants in RAB28 in cone-rod dystrophy'

This paper reports 4 novel variants in the RAB28 gene in 5 cases (from 4 families) with autosomal recessive cone-rod dystrophy, and provides evidence, primarily from protein conformation modelling, for their pathogenicity. As one of the rarer causes of cone-rod dystrophy, this confirmation of the involvement of RAB28 is helpful, and should assist with interpretation of any future novel variants in RAB28 found in diagnostic gene panel tests.

We are glad that this Reviewer found merit in our work. We have addressed all the points raised as follows.

The science and the writing of the paper are to a high standard, and revision requires only a few minor improvements to the English, and to the counting of the variants, plus a suggested improvement in the labelling of the colour diagrams.  These are :

  1. Abstract lines 39-41, and Introduction lines 82-84, and Discussion lines 444-445.  Four probands should have 8 affected alleles.  The listing of these here (i,ii,iii) suggests there are 9 affected alleles, which is confusing.  In fact there were 3 novel variants in homozygosis, but one of these is the same one as the splice site variant (Case 1), so that cannot be listed as a separate variant.  The authors must amend this sentence accordingly.  Also in this sentence the authors must make it clear that the nonsense variant in compound heterozygosity with the novel missense variant is not novel.  I would suggest rephrasing as ‘a novel missense and a previously reported nonsense variant in compound heterozygosis.’

 We agree with this Reviewer that the listing of the novel variants in our original manuscript could be misleading. We therefore rephrased lines 39-42 as follows: “By sequencing the four probands, we identified: a novel homozygous splicing variant; two novel nonsense variants in homozygosis; a novel missense variant in compound heterozygous state with a previously reported nonsense variant.”

  1. Introduction lines 82-84 . See Point 1. Above.

Lines 88-91 were rephrased as follows: “We identified two novel homozygous nonsense variants, a novel homozygous splicing variant and a novel missense variant in compound heterozygosis with a previously reported nonsense variant.”

  1. Discussion lines 444-445. See Point 1. Above

Lines 470-472 were rephrased as follows: “ … and identified two novel homozygous nonsense variants, a novel homozygous splicing variant, and a novel missense variant in compound heterozygosis with a previously reported variant.”

  1. Intro. Line 69. Better as : ‘A study on a murine model…’

 Line 75 was revised as requested.

  1. Intro. Line 71. Better as : ‘…proposed as the possible…’

 Line 76 was revised as requested.

  1. Intro. line 69-71. Also line 428-431. This whole sentence needs breaking up as it is unclear which aspects are observed and which are proposed.  This could be done by a comma between  ‘…phagocytosis, and….’  (if that is the right place to split), or perhaps better by making 2 sentences.  Ie.  ‘…shedding and phagocytosis.  An impaired membrane shedding at….’

To improve clarity, the sentence was split and lines 75-78 were rephrased as follows: “A study on a murine model showed that Rab-28 plays an essential role in cone-specific disc shedding and phagocytosis. This led to the proposition of an impaired membrane shedding at the distal Cone Outer Segment (COS) and/or failed phagocytosis by the RPE as the possible pathogenetic mechanism for CORD [9]”

  1. Res. 2.3 line 195-196. Better as : ‘…It can be noted from OCT imaging that there is a reduction…’

Line 206 was revised as requested.

  1. Re. 2.4 line 217. Better as : ‘…decreased progressively over the next ~34 years…’ 

Lines 227-228 were revised as requested.

  1. Res.2.4 line 224. ‘…novel…’ may be better than ‘…new…’

Line 234 was revised as requested.

  1. Res. 2.5 line 252 ‘…novel…’ may be better than ‘…new…’

 Line 262 was revised as requested.

  1. Res.2.6 line 271 ‘…peculiar…’ is not the correct word to use here. Do the authors mean : ‘characteristic’, ‘specific’, ‘individual’, or ‘different’ ?

We thank the Reviewer for suggesting a more appropriate word for “peculiar”, which was replaced with “specific” in line 282.

  1. Res.2.6 line 273 ‘…is composed of..’ rather than ‘…is composed by…’

Line 283 was revised as requested.

  1. Res.2.6 line 277 ‘…five α-helices (named α1 to α5) named an α/β fold…’ ; the repetition of ‘named’ is confusing. This might be better written as : ‘…five α-helices (named α1 to α5) forming an α/β fold…’ 

Line 287 was revised as requested.

  1. Res.2.6 line 278 ‘…substituted with…’ would read better as : ‘…substituted by…’

Line 289 was revised as requested.

  1. Fig7 line 308. Better here to start the sentence with:  ‘A) This cartoon shows…..

We prefer to keep the original sentence in the caption as “cartoon” is the technical definition of the representation used to evaluate the three-dimensional structure of the protein.

  1. Fig.7 line 308-313. It would be much better here to use a colour key in colour, rather than describing the colours. Can this be part of the figure itself; if necessary by making 7A a figure on its own ? 

We agree with this reviewer that the representation of each secondary structure element should be made clearer. To this purpose we added an inset showing the structural elements colored according to the structure and modified the caption (lines 321-323) as follows: “Inset shows the coloring scheme for secondary structure elements.” and deleted the description of each part in colour.

  1. Fig.7 line 313-318. For clarity of reading, could B),C) & D) all start on a new line ?

Lines 318, 326 and 330 were revised as requested.

  1. Molec. Modelling line 324. ‘…Switch 1 region resulted considerably more flexible in both…’   It is difficult to follow what this sentence means.  Do the authors mean : ‘‘…Switch 1 region was found to be considerably more flexible in both…’   Please edit as appropriate.

We rephrased line 336 as suggested by this Reviewer.

  1. Molec. Modelling line 329. Better to say: ‘…It is noteworthy that while the enhanced…’

Line 342 was revised as requested.

  1. Molec.Modelling line 340. ‘…as the cation resulted less tightly bound…’ Do the authors mean: ‘…as the cation remained less tightly bound…’   or ‘…as the cation was less tightly bound…’  Please change as appropriate. 

Line 352 was rephrased as follows “… as the cation was found to be less tightly bound in the inactive form…”

  1. Fig.8. line 396-412 Please label the GTP in the diagram. Again here, as with Fig7, it would seem better to use a colour key in colour, rather than describing the colours

We agree with this Reviewer that Figure 8 should be improved in terms of clarity and labeling. We therefore split panel A and panels B,C and D in two figures. Figure 8 now represents separately the structure of the four superimposed structure of panel 8A, with the following caption: “Figure 8. A) Comparison of the three-dimensional structure of A) GTP-bound WT (black), B) GTP-bound T26N (blue), C) G3D-bound WT (red) and D) G3D-bound T26N (green) Rab-28 after 500 ns MD simulations. Protein structure is shown as a ribbon connecting Cα atoms, Mg2+-ion is represented as a sphere colored with the same color as the protein structure, GTP/G3D are represented as sticks with N atoms in blue, O atoms in red and P atoms in orange and labelled. The switch 1 region of each state is highlighted with a lighter hue of the protein structure color and labelled.”

Panels B, C and D are now panels A, B and C of Figure 9. Figure caption was adapted as follows: “Figure 9. A) Three-dimensional structure of GTP-bound WT Rab-28, protein structure is represented as an orange cartoon, Mg2+-ion is shown as an orange sphere, GTP is displayed as magenta sticks, the molecular surface of each element is shown in transparency and colored accordingly. B) Three-dimensional structure of GTP-bound p.(Arg137*) Rab-28, protein structure is represented as a teal cartoon, Mg2+-ion is shown as a teal sphere, GTP is displayed as magenta sticks, the molecular surface of each element is shown in transparency and colored accordingly. C) Three-dimensional structure of GTP-bound p.(Trp107*) Rab-28, protein structure is represented as a light blue cartoon, Mg2+-ion is shown as a light blue sphere, GTP is displayed as magenta sticks, the molecular surface of each element is shown in transparency and colored accordingly. The GTP molecule shown in panels B) and C) is merely for representation purposes, as both truncated forms would not allow the correct folding of the binding pocket.”

  1. Disc. Line 428-431. See Point 6. Above (this sentence is repeated from Intro. Line 69-71. 

Lines 470-473 were rephrased as follows: “ … two novel homozygous nonsense variants, a novel homozygous splicing variant, and a novel missense variant in compound heterozygosis with a previously reported variant.

  1. Disc. Line 437-438. This sentence would be better written as : ‘The phenotype of these patients presented features in common, showing a…’

 Lines 462-463 were revised as requested.

  1. Disc. Line 440-442. The reading of this sentence is ambiguous. Do the authors mean: ‘Only in the case of the p.(Ser23Phe) variant could a three-dimensional structure of the wild type and mutated RAB28 proteins be predicted, and this suggested impaired ligand binding [4]’ ;  or do they mean: ‘Only in the case of the p.(Ser23Phe) variant did a predicted three-dimensional structure of the wild type and mutated RAB28 proteins suggest impaired ligand binding [4]. ?

We agree with the Reviewer, so we rephrased the sentence in lines 465-468 to clarify the ambiguity as follows: “The comparison of the predicted three-dimensional structure of the p.(Ser23Phe) variant with wild type RAB28 protein suggested impaired ligand binding [4]. “

  1. Disc. Line 450. Better to write: ‘In FAF imaging all patients presented a…’ 

Line 478 were revised as requested.

  1. Disc. Line 452-453 Please add in a comma: ‘…and rod-cone dystrophy [21,22], and is a non-specific…’

 Line 481 were revised as requested.

  1. Disc. Line 463-464. Which of the sister or brother was the proband ? Here it suggests the sister, whereas in Table 2 it is the brother who is listed as the proband. Please make these consistent.

 We agree with the reviewer that in our original manuscript it was not clear who was the proband and have changed two sentences as follows:

Line 199, “The 38-year-old subject is the sister of patient 2”.

Lines 491-493, “Foveal atrophy was less pronounced in the proband’s older sister resulting in a more preserved visual acuity; similarly, visual field was less altered in the female subject, who also showed a later onset of the disease compared with her brother” in Discussion.

  1. Disc. Line 464. ‘…disease compared with her brother…’ would seem better than : ‘…disease in respect to the brother…’.

Line 493 was revised as requested.

  1. Disc. Line 468. Better with a comma: ‘…CORD, while…’

Line 497 was revised as requested.

  1. Disc. Line 473. ‘…in contrast to the mouse…’ would be better than:  ‘…differently from the mouse…’
    Line 502 was revised as requested.

  1. Disc. Line 496. Better as : ‘…understanding of the role of…’ 

Line 525 was revised as requested.

  1. Abbreviations line 578. Please list the abbreviations in alphabetical order.

Abbreviations list was sorted in alphabetical order.

Round 2

Reviewer 1 Report

Thanks to authors for making some editing and improvement in the content, but unfortunately these modifications are not enough to support the claims. Hence, this manuscript in its present form is not suitable for publication in this journal.

Author Response

We regret that this Reviewer thinks that the amendments to the original manuscript are not sufficient and concludes that the manuscript in its present form is not suitable for publication in this journal.

Since we have addressed all the points raised by this Reviewer, we do not understand the reason of such generic comment, which sounds rather superficial and therefore useless if not supported by clear statements.